# Metagenomic next generation sequencing of plasma RNA for diagnosis of unexplained, acute febrile illness in Uganda

Abraham J. Kandathil[1], Paul W. Blair[1,2], Jennifer Lu[3,4,5],
Raghavendran Anantharam[1], Kenneth Kobba[6], Matthew L. Robinson[1], Sultanah Alharthi[2],
Edgar C. Ndawula[6], J. Stephen Dumler[2,5], Francis Kakooza[6], Mohammed Lamorde[6], David
L. Thomas[1], Steven L. Salzberg[3,4,7], Yukari C. Manabe[1] *

1 Division of Infectious Diseases, Johns Hopkins School of Medicine, Baltimore, Maryland, United States of America, 2 Department of Pathology, Uniformed Services University, Bethesda, Maryland, United States of America, 3 Center for Computational Biology, Johns Hopkins University, Baltimore, Maryland, United States of America, 4 Department of Biomedical Engineering, Johns Hopkins University, Baltimore, United States of America, 5 Department of Pathology, Johns Hopkins School of Medicine, Baltimore, Maryland, United States of America, 6 Infectious Diseases Institute, Makerere University, Kampala, Uganda, 7 Departments of Computer Science and Biostatistics, Johns Hopkins University, Baltimore, Maryland, United States of America

☯ These authors contributed equally to this work.
* ymanabe@jhmi.edu

## Abstract

Metagenomic next generation metagenomic sequencing (mNGS) has proven to be a useful tool in the diagnosis and identification of novel human pathogens and pathogens not identified on routine clinical microbiologic tests. In this study, we applied mNGS to characterize plasma RNA isolated from 42 study participants with unexplained acute febrile illness (AFI) admitted to tertiary referral hospitals in Mubende and Arua, Uganda. Study participants were selected based on clinical criteria suggestive of viral infection (i.e., thrombocytopenia, leukopenia). The study population had a median age of 28 years (IQR:24 to 38.5) and median platelet count of 114 x$10^3$ cells/mm$^3$ (IQR:66,500 to 189,800). An average of 25 million 100 bp reads were generated per sample. We identified strong signals from diverse virus, bacteria, fungi, or parasites in 10 (23.8%) of the study participants. These included well recognized pathogens like *Helicobacter pylori*, human herpes virus-8, *Plasmodium falciparum*, *Neisseria gonorrhoeae*, and *Rickettsia conorii*. We further confirmed *Rickettsia conorii* infection, the cause of Mediterranean Spotted Fever (MSF), using PCR assays and Sanger sequencing. mNGS was a useful addition for detection of otherwise undetected pathogens and well-recognized non-pathogens. This is the first report to describe the molecular confirmation of a hospitalized case of MSF in sub-Saharan Africa (SSA). Further studies are needed to determine the utility of mNGS for disease surveillance in similar settings.

**Data Availability Statement:** The metagenomic dataset generated and analyzed during the current study have been deposited in Sequence Read Archive with the accession code PRJNA1115716.

**Funding:** This study was funded by the CAPA-CT II project (RIA2018EF-083 to ML and YCM) which is part of the EDCTP2 program supported by the European Union. Salary support was provided by the U.S. National Institutes of Health under grants R35-GM130151 for S.L.S., National Institute on Drug Abuse grants R01DA048063 for D.T., R21DA053145 for A.J.K., National Institute for Biomedical Imaging and Bioengineering U54EB007958 and Fogarty International Center 5D43TW009771 for YCM. The funders had no role in study design, data collection and analysis, decision to publish, or preparation of the manuscript.

**Competing interests:** The authors have declared that no competing interests exist.

## Author summary

Unbiased molecular approaches like metagenomic sequencing have improved our ability to identify not only novel microbes but also known microbes in new settings. We report the results of a metagenomic next generation sequencing approach to identify viral and cell free plasma RNA from a curated panel of 42 unexplained, acute febrile, hospitalized study participants from tertiary referral hospitals in Mubende and Arua, Uganda. In ten study participants, metagenomic sequencing allowed us to identify pathogens including *Helicobacter pylori*, and *Rickettsia conorii* that were missed on routine clinical microbiologic testing. Sequence-specific targeted PCR assays and Sanger sequencing confirmed *Rickettsia conorii* infection in the first hospitalized case of Mediterranean Spotted Fever diagnosed in sub-Saharan Africa. Using appropriate controls, we observed metagenomic sequencing to be unable to consistently detect microbial sequences when plasma circulation levels were below 10,000 copies/mL. These results highlight the need for more studies to determine the utility of metagenomic next generation sequencing approaches for disease diagnosis and surveillance in similar settings.

## Introduction

The threat of emerging infections, limitations of target-specific diagnostics, decreasing costs, and improving bioinformatics have contributed to broader interest in unbiased metagenomic next-generation sequencing (mNGS) as a tool to identify the cause of undiagnosed infections [1–4]. This approach has facilitated rapid identification of novel outbreak pathogens which in turn can accelerate development of pharmacological and epidemiologic interventions [5,6]. However, while the agnostic mNGS approach to pathogen identification can be useful to overcome limitations of standard microbiologic diagnostics, questions remain about the optimal approach, patient use case, and net diagnostic gain. Careful sample selection, laboratory protocols, and computational analysis are needed to decrease misinterpretation of results from contamination, transient viremia, or nonpathogenic members of the human virome [7,8]. Additionally, careful selection of samples, knowing the clinical context, and/or performing additional confirmatory tests are important factors for optimizing the value of mNGS.

Some acute febrile illness (AFI) is caused by infections like malaria that can be diagnosed at the point-of-care. However, many cases of AFI remain undiagnosed, either because of limitations in diagnostic testing or the emergence of new pathogens or re-emergence of a known pathogen in an unexpected setting [9].We sought to evaluate the potential for RNA mNGS to identify possible microbial causes of unexplained acute febrile illness (AFI) in East Africa. Plasma metagenomic sequencing was performed on a curated panel of study participants selected from an AFI cohort based in Uganda [10]. We used RNA rather than DNA sequencing to enable the detection of both plasma RNA viremia and highly expressed microbial RNA from emerging and reemerging human infections in Rift Valley of East and Central Africa [11–13]. This approach resulted in the identification and molecular confirmation of a bacterium, *Rickettsia conorii*, as the cause of locally acquired Mediterranean spotted fever (MSF), causing severe illness, requiring hospitalization in sub-Saharan Africa (SSA).

## Study design and methods

### Ethics statement

This study was conducted in compliance with the Declaration of Helsinki and Good Clinical Practice Guidelines. The study and informed consent process were reviewed and approved by

the Joint Clinical Research Centre (JCRC) Research Ethics Committee (JC1518) and the Uganda National Council for Science and Technology (UNCST), HS 371ES, and Johns Hopkins University School of Medicine Internal Review Board (IRB00176961). All participants signed written informed consents prior to study procedures. All samples were de-identified prior to laboratory testing.

## Study participants

This cross-sectional study sequenced plasma from individuals admitted with acute febrile illness to tertiary referral hospitals in Mubende and Arua, Uganda between 2019 to 2020[10]. In brief, adults ($\geq$18 years of age) in the emergency or outpatient departments to be admitted with a recorded fever ($\geq$ 38.0˚C) or clinical history of fever within the past 48 hours and had symptom onset within the past 2 weeks were eligible for enrollment in the previously described parent cohort. After consent, study personnel collected baseline physiologic parameters, plasma and serum for standardized clinical diagnostic tests including HIV testing with consent (Determine HIV1/2: Abbott, OK, USA; Stat-Pak, Chembio Diagnostics Systems, Inc, NY, USA; SD Bioline: Gyeonggi-do, Republic of Korea), malaria rapid test (CareStart malaria Pf HRP2 antigen RDT), and thick malaria smears as previously described [10]. Participants were followed at one-month through in-person or telephone visits for blood collection and for determining survival. The participants in this mNGS study panel were selected based on i) clinical indication of viral infection (i.e., thrombocytopenia, leukopenia) and ii) absence of a validated laboratory diagnosis to explain the clinical presentation. As one of the aims of this study was to identify emerging and remerging human infections, we prioritized sequencing of samples from patients without HIV as these patients often present with febrile illness caused by well-described opportunistic infections [14].

**Sample Preparation for next generation sequencing (NGS):** i) *Nucleic Acid Extraction*: Extraction was performed on 200 μL of plasma using ZR-Duet DNA/RNA miniprep kit (Zymo Research, CA, USA) as previously described [15]. Pre-extraction steps done to minimize non-viral sequences included centrifugation of plasma at 1600g for 15 minutes at 4˚C followed by filtration of the supernatant using 0.2 μM syringe filters (Whatman, Amersham Place, UK). This extraction protocol allowed for separate elution of the RNA fractions, thereby reducing DNA contamination.

ii) *Metagenomic next generation sequencing (mNGS)*: The NGS library constructions were performed using Ovation SoLo RNA-Seq (Tecan Genomics, Inc., CA, USA) library preparation kit. The kit was selected based on its ability to generate libraries from low biomass samples like plasma. Additionally, during library preparation extra amplification cycles were incorporated to increase the final concentration of the cDNA libraries. Qualitative assessment of each library was done using a 2100 Bioanalyzer (Agilent Technologies, CA, USA) to visualize fragment size distribution and identify presence of non-specific DNA fragments that could potentially affect read quality. Although the average library size ranged from 300–350 bp, samples that had peaks prior to the library range (likely due to primer dimer or adaptor dimers) were also taken for sequencing as these were smaller in proportion to the library peak. In our experience these do not affect read quality. This was followed by quantitative assessment using the Kappa PCR library kit (Roche, IN, USA). Samples were pooled in batches to yield a final concentration of 4nM for sequencing on a NovaSeq 6000 (Illumina, Inc., CA, USA). Required volumes of each sample library was determined using the pooling calculator made available at https://support.illumina.com/help/pooling-calculator/pooling-calculator.htm. Sequencing of pooled samples was done as technical replicates (two lanes) on a single flow cell using high output mode with a read length of 2X100 bp and 500 million reads/lane.

iii) *Controls*: Both positive and negative controls were included as extraction and sequencing controls. Positive controls included a) two plasma specimens with known clinical and laboratory diagnosis (*Plasmodium falciparum* and human immunodeficiency virus [HIV]) and b) AcroMetrix Multi-Analyte (Thermo Fisher Scientific, MA, USA). The multianalyte contained SARS-CoV-2, Flu A/B, and RSV A/B at a concentration of 10,000 copies/mL. Water was used as a negative control. The controls were processed identically to the study samples.

iv) *Taxonomic classification of reads*: All sequencing reads were aligned to the human genome T2T-CHM13[16] using Bowtie 2[17]. The remaining non-human reads were classified by KrakenUniq [18] against a customized database built from all complete bacterial, archaeal, and viral genomes from NCBI RefSeq, a collection of 256 eukaryotic pathogens from EuPathDB, UniVec laboratory vectors from NCBI (https://www.ncbi.nlm.nih.gov/tools/vecscreen/univec/) and the GRCh38 human genome. The KrakenUniq database includes 46,711 bacterial genomes (5981 species), 13,011 viral genomes (9905 species), and 604 archaeal genomes (295 species), and the full database can be downloaded from https://benlangmead.github.io/aws-indexes/k2

v) Computational analysis: The KrakenUniq reports for all samples were analyzed for potential infectious agents using Pavian [19] to compare read counts between all samples and the sample controls (S1 Table). Using Pavian's z-score, we isolated all species with significantly higher read counts than most other samples or controls, considering these species as potential microbial infections. Reads for any potential microbial infection were extracted using Kraken-Tools[20] and realigned using Bowtie 2 to visualize genome coverage of the classified reads. We also looked for indications of false positives by identifying plots that showed localization of reads in only 1–2 regions. Additionally, KrakenUniq algorithm reports a "unique k-mer" count, i.e., reads that match low-complexity regions. e.g. a long (ATATAT. . .) repeat will have 2 k-mers while a 100bp read should have seventy 31-mers if they are all different. Hence, if KrakenUniq reports 100 reads but only 200 k-mers, that means all of those reads hit the same spot either due low complexity or a contaminant. The presence of the microbial species was considered valid only if the number of uniq k-mers were much greater than the number of reads. We primarily used the calculated z-score to identify potential calls, which were then further validated using both the k-mer counts and the genomic coverage by the reads. Clinical relevance was determined by a committee of three board-certified infectious diseases specialists.

vi) *Rickettsia PCR*: For confirmation of rickettsial infection, DNA was extracted from serum using QIAamp RNA Mini Kit (Qiagen, Venlo, Netherlands). PCR was performed using previously published methods targeting mRNA of spotted fever group *sca0* (OmpA gene), *R. conorii*-specific *sca0*, and typhus group 17-kDa outer membrane protein gene, and positives were called only if in duplicate [21,22]. The 127-basepair amplicon from *R. conorii*-specific *sca0* PCR plate was sent for Sanger sequencing and aligned to a reference genome using Snap-Gene software (GLS Biotech, San Diego, CA, USA).

## Results

The 42 study participants included participants hospitalized in Mubende, Uganda (n = 26) and in Arua, Uganda (n = 16). Participants had a median age of 28 years (IQR:24, 38.5) with 50% identifying as females (**Table 1**). Based on the study selection criteria, participants had leukopenia and/or thrombocytopenia with a median WBC and platelet count of 5.3 x10$^3$ cells/mm$^3$(IQR:3.6 to 9.1) and 114 x10$^3$ cells/mm$^3$ (IQR:66.5 to 189.8), respectively. Although we enriched for virions during RNA extraction, the protocol allowed us to simultaneously sequence cell free RNA. Hence, in addition to identification of RNA virome, we were also able to identify infections due to bacteria, fungi, parasites, and DNA viruses.

**Table 1. Baseline Demographic and Vital Physiological Parameters of the study panel.**

| Characteristic | Total (N = 42) | Arua RRH (N = 16) | Mubende RRH (N = 26) |
|---|---|---|---|
| Female sex—no. (%) | 21 (50.0) | 8 (50.0) | 13 (50.0) |
| Age—years, median (IQR) | 28.0 (24.0, 38.5) | 29.0 (24.0, 37.8) | 28.0 (24.0, 39.0) |
| HIV (%) | 2 (4.8) | 1 (5.9) | 1 (3.8) |
| Duration of symptoms—days, median (IQR) | 4.5 (3.0, 7.0) | 4.0 (3.0, 5.3) | 5.0 (3.0, 7.0) |
| **Physiological Parameters** (IQR) | | | |
| Heart rate (beats per minute) | 110.0 (98.3, 128.8) | 98.5 (90.5, 109.0) | 116.0 (101.2, 132.0) |
| Temperature (˚C) | 38.4 (38.1, 38.8) | 38.5 (38.0, 38.6) | 38.4 (38.1, 38.8) |
| Systolic blood pressure (mmHg) | 112.0 (100,119.0) | 113.5(99.0,119.0) | 110.5(101.0,120.0) |
| Diastolic blood pressure (mmHg) | 67.5 (59.3, 75.8) | 67.5 (61.3, 75.3) | 66.5 (59.3, 77.3) |
| Respiratory rate (breaths per minute) | 22.0 (18.0, 26.8) | 21.0 (17.3, 27.3) | 22.0 (18.0, 25.8) |
| Oxygen saturation (%) | 98.0 (96.0, 98.0) | 98.0 (97.8, 98.3) | 97.0 (95.3, 98.0) |
| Glasgow coma scale | 15.0 (15.0, 15.0) | 15.0 (15.0, 15.0) | 15.0 (15.0, 15.0) |
| qSOFA score≥ 2—no. (%) | 10 (23.8) | 4 (25.0) | 6 (23.1) |
| **Outcome** | | | |
| Died (%) | 10 (23.8) | 5 (31.3) | 5 (19.2) |
| Duration of hospitalization—days, median (IQR) | 2 (2.0, 4.5) | 3 (2.0, 4.8) | NA |

RNA sequencing of all samples yielded an average of 25 million 100 bp paired reads per sample. 83.5% of all reads were identified as human using Bowtie 2 and KrakenUniq. Removal of human sequences resulted in an average of 1.4 million reads per sample (**Table 2**).

**Table 2. KrakenUniq classified non-human read counts per sample.**

| Sample | Classified Non-Human Reads | Sample | Classified Non-Human Reads |
|---|---|---|---|
| AFI-1 | 700,577 | AFI-29 | 630,375 |
| AFI-6 | 1,691,350 | AFI-30 | 3,192,999 |
| AFI-7 | 862,091 | AFI-31 | 547,573 |
| AFI-8 | 470,110 | AFI-32 | 12,408,653 |
| AFI-9 | 928,205 | AFI-33 | 3,561,562 |
| AFI-10 | 387,198 | AFI-34 | 793,952 |
| AFI-11 | 643,348 | AFI-35 | 1,630,684 |
| AFI-12 | 552,810 | AFI-36 | 1,520,607 |
| AFI-13 | 258,230 | AFI-37 | 3,595,947 |
| AFI-14 | 45,475 | AFI-38 | 1,422,173 |
| AFI-15 | 182,264 | AFI-39 | 819,044 |
| AFI-16 | 3,148 | AFI-40 | 669,243 |
| AFI-17 | 1,485,407 | AFI-41 | 2,088,918 |
| AFI-18 | 205,858 | AFI-42 | 1,419,928 |
| AFI-19 | 943,131 | AFI-43 | 179,161 |
| AFI-20 | 2,836,251 | AFI-44 | 424,884 |
| AFI-21 | 490,829 | AFI-45 | 38,618 |
| AFI-22 | 872,502 | AFI-46 | 186,514 |
| AFI-24 | 445,745 | AFI-47 | 1,840,597 |
| AFI-25 | 121,075 | AFI-48 | 2,817,500 |
| AFI-26 | 529,889 | AFI-49 | 3,060,312 |
| AFI-27 | 395,909 | NTC | 993,408 |
| AFI-28 | 2,900,233 | ACROMETRIX | 5,944,038 |

This study sequenced 42 participants, AcroMetrix control, No Template Control (NTC) and 2 with known clinical and laboratory diagnosis (AFI-6 and AFI-12).

Analysis of KrakenUniq read classifications revealed the expected *Plasmodium falciparum* species signal in the positive control (S2 Table). However, the protocol failed to identify HIV-1 present at a concentration <4 log $_{10}$ copies/ml. This is consistent with our previous observation that metagenomic sequencing can give false negative results for viruses at concentrations <4 log $_{10}$ copies/mL [15]. Additionally, the KrakenUniq protocol and Pavian analysis identified SARS-CoV-2, Flu A/B, and RSV A/B, all of which were present at a concentration of 4 log $_{10}$ copies/mL in the AcroMetrix sample.

Further analysis of the KrakenUniq classifications identified 10 microbes among the 42 study participants with significant Pavian z-scores (Tables 3 and S2). These included organisms that are always considered pathogens when isolated from human specimens (*N. gonorrhoeae*, *H. pylori*, *P. falciparum*, and *R. conorii*). We also identified microbes that can be normal flora or pathogenic in certain settings (*Candida parapsilosis*, human herpes virus-8 [HHV-8], *Human mastadenovirus C*, and *Staphylococcus haemolyticus*) [23–25]. *Pegivirus hominis*(also known as pegivirus-C), a commonly observed non-pathogen in blood, was also detected in two participants[26]. **Fig 1** displays the read counts for these 10 microbes across all samples.

## Cases with clinically relevant mNGS results

Among the four cases adjudicated to have had clinically relevant results, the pathogens were not identified on protocolized routine clinical microbiologic testing (**Table 2**). First, *N. gonorrhoeae* sequences were found in a 27-year-old woman with abdominal pain and fever. Routine testing revealed a positive anti-treponemal antibody syphilis test. The patient was diagnosed with syphilis and was treated with intramuscular penicillin. This patient received intravenous ceftriaxone (active against *N. gonorrhoeae*); standard blood cultures were negative. Ceftriaxone was given empirically. Second, *H. pylori* sequences were detected in a 40-year-old woman with fever and abdominal pain who was clinically diagnosed with gastritis and ulcers. The clinical course was consistent with *H. pylori* infection, but routine access to *H. pylori* testing was not available in this setting. Third, *P. falciparum* sequences were found in a 38-year-old man with fever, altered mental status, seizures who died during admission. Rapid diagnostic and smear testing for malaria were negative.

## Mediterranean spotted fever (R. conorii)

The fourth clinically relevant case is a 39-year-old man with no past medical history who presented 7 days after symptom onset with abdominal pain, altered mental status, a generalized rash, and rigors. He was previously seen in a clinic and prescribed antimalarials (artesunate and quinine tablets) which he took for 6 days without improvement. He was a crop and livestock farmer from Mubende. He reported mosquito bites but no known tick bites. He had goats and chickens around his household. He had a fever of 40.2°C, confusion (Glasgow coma score 14), heart rate 119 bpm, blood pressure of 81/47 mmHg, respiratory rate of 38 breaths per minutes, and oxygen saturation of 87%. On exam he was toxic appearing, had nuchal rigidity, crackles on lung auscultation, tachycardia on heart auscultation, abdominal tenderness, and a papular rash. His white blood count was 4.7 x10$^3$ cells/μL, hemoglobin 16.9 g/dL, platelet count 89 x10$^3$ cells/μL, AST 92 U/L and ALT 39 U/L. Routine microbiology laboratory results were negative for HIV, malaria, hepatitis A and hepatitis B, and blood cultures were without growth. He was admitted with a diagnosis of meningitis and pneumonia and treated with clindamycin, levofloxacin, and amikacin. He remained febrile at 39.4°C, but heart rate improved to 90 bpm, respiratory rate 22 breaths per minutes, and blood pressure 120/80 mmHg with

**Table 3. Metagenomic next generation sequencing diagnosis and clinical presentations.**

| Study ID (site) | Age (yrs) | Sex | Medical History | Diagnosis | | |
|---|---|---|---|---|---|---|
| | | | | *Clinical* | *mNGS* | *Relevance* |
| AFI 1 (Arua) | 27 | F | Two days of abdominal pain, fever to 38.1°C, who was HIV-negative, but RPR-positive with a WBC elevated at 14.4 K/mm$^3$ alive at 28 days, discharged after receiving IV ceftriaxone. | Sepsis of unknown source and syphilis | *N. gonorrhoeae* | + |
| AFI 10 (Mubende) | 20 | F | Fever of 38.2°C, tachycardic (HR 142 bpm), hypotensive (90/62 mmHg) with generalized systemic symptoms (fatigue, headache, myalgias, arthralgias, nausea, vomiting, rigors) with evidence of urinary tract infection, jaundice, and splenomegaly, HIV-negative, treated with ciprofloxacin and metronidazole discharged after 2 days unable to reach at 28 days. | Urinary tract infection | *Staphylococcus hemolyticus* | +/- |
| AFI 13 (Arua) | 30 | F | Fever 38.0°C, HIV+ (CD4 118) on antiretroviral therapy who presented with 7 days of diarrhea, headache, nausea, vomiting for 7 days, anemic (hgb 9 g/dl), low platelets (25 K/mm$^3$), decreased kidney function (creatinine 1.5 mg/dL) and mildly elevated liver enzymes (ALT 65 U/L and AST 101 U/L) who died while hospitalized. | Sepsis of unknown source and gastroenteritis | Human Herpesvirus-8 | +/- |
| AFI 19 (Arua) | 25 | M | Fever 38.8°C presents with chest pain and dyspnea (anorexia, cough, headache) for 7 days who had been seen previously at a clinic, HR 130 bpm, RR 25 bpm, SpO2 68%, toxic appearing with crackles, urinalysis with pyuria, WBC 9.1 K/mm$^3$, platelets 241 K/mm$^3$, hgb 10.7 g/dL, HIV-negative, treated with cefotaxime, died while hospitalized. | Pneumonia and urinary tract infection | *Candida parapsilosis* | +/- |
| AFI 21 (Mubende) | 40 | F | Fever 38.4°C presents with abdominal pain, anorexia, arthralgia, dysuria, fatigue, headache, rigors for 4 days, toxic appearing with stable vital signs, tender abdomen, diagnosed with sepsis and gastritis, WBC 4.6 K/mm$^3$, platelets 157 K/mm$^3$, treated with metronidazole and cefotaxime, alive at follow-up | Sepsis and gastritis | *Helicobacter pylori* | + |
| AFI 24 (Mubende) | 39 | M | Farmer with direct contact with goats and chickens with fever 40.2°C, admitted with abdominal pain, altered mental status, anorexia, diarrhea, fatigue, headache, rash and rigors for 7 days HR 119 bpm, hypotensive 81/47 mmHg, RR 38 bpm, confused, toxic, nuchal rigidity, crackles, tender abdomen, papular rash, WBC 4.7 K/mm$^3$, plt 89 K/mm$^3$, treated with clindamycin, levofloxacin, amikacin and improved after 1 day, clindamycin and kanamycin on day 2. Alive at day 28 | Meningitis and pneumonia | *Rickettsia conorii* | + |
| AFI 27 (Arua) | 38 | M | Crop farmer with fever 38.6°C, change mental status, headache. Recurrent seizures for 5 days, 154/101 mmgHg, SpO$_2$ 93%, GCS 5, toxic, not opening eyes, nuchal rigidity, malaria smear and RDT negative, WBC 8.2 K/mm$^3$, hgb 10.6 g/dL, platelets 102 K/mm$^3$, AST 306 U/L, ALT 93 U/L, treated with ceftriaxone, metronidazole, patient died while hospitalized. | *Possible meningitis* | *Plasmodium falciparum* | + |
| AFI 28 (Mubende) | 23 | F | Teacher with 5 days of shortness of breath, cough, headache, chest pain, abdominal pain, nausea, vomiting presenting with fever of 38.7°C, heart rate 100 bpm, 210/162 mmHg, SpO$_2$ 96% and lung crackles on exam. WBC 5.1 K/mm$^3$, hgb 13.1 g/dL, creatinine 6 mg/dL. Chest X-ray revealed right base opacities. | *Pneumonia* | human mastadenovirus C | - |
| AFI 35 (Arua) | 19 | F | Fever 38.0°C with abdominal pain, cough, diarrhea, headache, nausea for 4 days, previously seen at clinic and received ACT, HR 107 bpm, 113/67 mmHg, RR 27 bpm, conjunctival pallor, splenomegaly, hgb 3.8 g/dL, platelet count 124 K/mm$^3$, WBC 6.4 K/mm$^3$, treated with ciprofloxacin, could not reach at 28 days. | Gastroenteritis and severe anemia | *Pegivirus hominis/ pegivirus-C* | - |
| AFI 47 (Mubende) | 40 | M | Fever 38.1°C, abdominal pain, anorexia, arthralgias, bleeding, dyspnea, fatigue, headache, myalgias, nausea, sore throat for 5 days, toxic, conjunctival pallor, crackles, tachycardia, hepatosplenomegaly, macular rash, Chest X-ray with infiltrates, hemoglobin 5.1 g/dL, platelet count 550 K/mm$^3$, white blood count 16 K/mm$^3$, treated with ceftriaxone, alive at 28 days. | Pneumonia and urinary tract infection | *Pegivirus hominis/ pegivirus-C* | - |

Note: mNGS: metagenomic next generation sequencing; ALT: alanine transaminase; AST: aspartate transaminase; bpm: beats or breaths per minute; F: female; GCS: Glasgow coma scale; HIV: human immunodeficiency virus; hgb: hemoglobin; HR: heart rate; M:male; RPR: rapid plasma reagin, RR: respiratory rate; WBC: white blood count

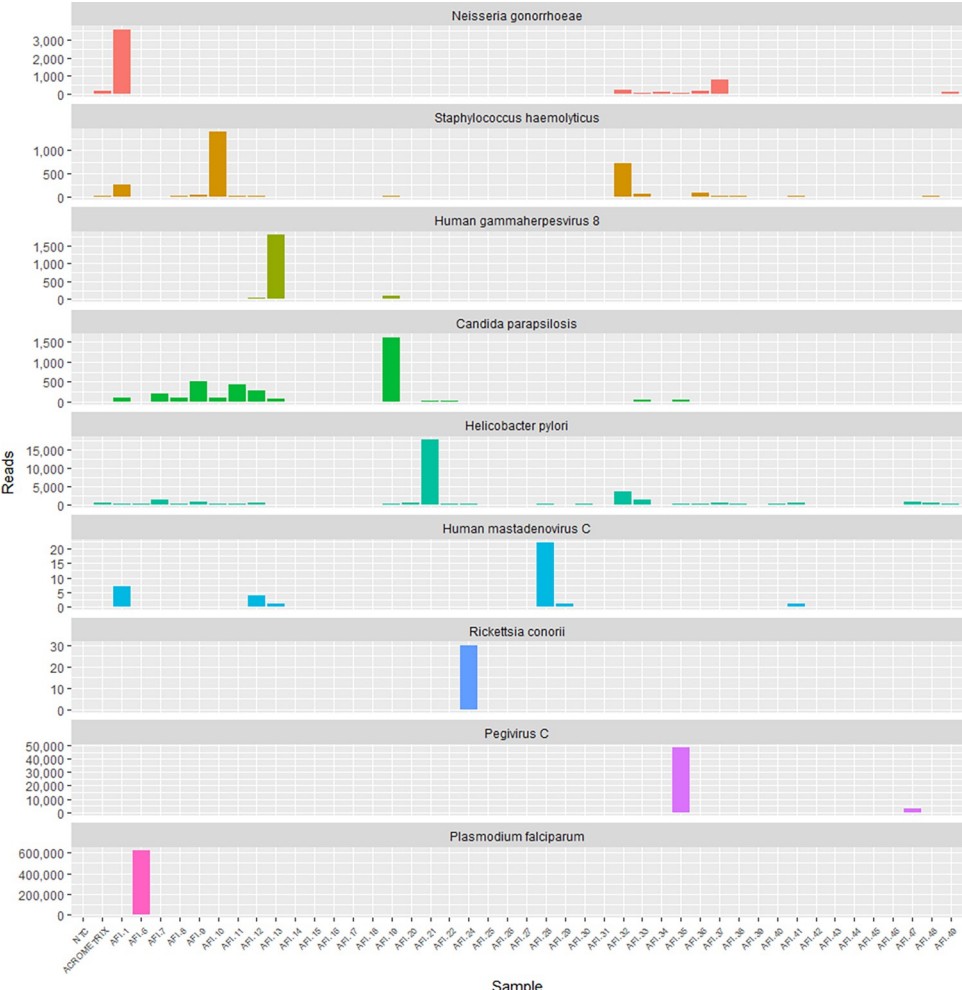

**Fig 1. KrakenUniq read classifications identified microbes across 42 study samples.** KrakenUniq read classifications for 9 identified microbes across 42 study samples are shown. The identified reads were absent in negative control (NTC), and the AcroMetrix control. Each of these microbes had significantly higher total read counts in 1–2 samples as compared to the remaining samples and controls, as determined by the Pavian z-score.

97% oxygen saturation. He was discharged by the following day. The participant was alive at 28 days per phone follow-up.

The baseline enrollment plasma sample was tested using real-time PCR and was positive for the spotted fever group *sca0* (*ompA*) target at a mean cycle threshold of 32.7. Typhus group 17-kDa PCR was negative. Serum samples were positive for *R. conorii*-specific *sca0* primers at a mean $C_t$ of 33.6 in three of four plasma duplicates. The amplicon was Sanger sequenced and had a 100% match with a *R. conorii* Malish 7 strain reference sequence.

## Discussion

In a hospitalized, acute febrile illness cohort in Uganda, RNA-based mNGS identified clinically relevant pathogens in a subset of participants without a microbiologic diagnosis. This report underscores the potential and limitations of mNGS for diagnostics and pathogen discovery. The findings of this study also include the first report of a molecularly confirmed locally acquired *R. conorii* infection leading to hospitalization in SSA [27,28].

The use of mNGS allowed identification of pathogens missed by standardized diagnostic tools. The pathogens identified included *R. conorii*, *H. pylori*, *N. gonorrhoeae*, and *P. falciparum*. There are few descriptions of rickettsial illness among those hospitalized in SSA with cases primarily limited to returning travelers. In the absence of surveillance programs and diagnostics, species causing rickettsial illness among those hospitalized with AFI are also largely unknown. Notably, the MSF participant was initially misdiagnosed with malaria and developed progressive illness including signs and symptoms of meningitis and unstable vital signs. Fortunately, empiric treatment with a fluoroquinolone, to which *R. conorii* is susceptible, resulted in dramatic recovery and swift discharge. This case report along with recent evidence of MSF circulation in tick pools throughout Uganda provides strong clinical evidence that MSF should be considered in patients with severe non-malarial illness leading to hospitalization in Uganda [29]. Availability and application of NGS for clinical purposes could provide valuable pathogen agnostic approaches to diagnosis after first-line testing is exhausted and to inform regional diagnostic needs. As with the identification of *R conorii* infection, detection of *H. pylori* sequences in the plasma of a patient with epigastric pain and ulcers was unexpected and clinically meaningful. This finding has been previously described and likely represents a real pathogen [30]. While clinically 'expected,' mNGS was the sole evidence of *N. gonorrhoeae* and malaria infections despite routine diagnostic testing in two other patients. Molecular detection of *P. falciparum* with a negative blood smear does not implicate malaria as the primary cause of acute febrile illness. However, identification of *P. falciparum* could provide context as an infection that could contribute to anemia, splenomegaly, or non-typhoidal infection [31].

Our findings also reflect some of the limitations of mNGS. mNGS is not widely available clinically and computational analysis require specialized bioinformatics tools for precise identification of microbial reads. As with existing blood cultures and even some clinical PCR diagnostics, microbes unlikely to be clinically significant (e.g., coagulase-negative staphylococcus) may be identified and lead to incorrect diagnoses or treatments. Hence, like routine interpretation of laboratory results, careful incorporation of clinical interpretation is needed. We also failed to identify evidence of microbial infection in most persons and failed to detect HIV RNA in one positive control, underscoring the limited sensitivity of the approach in detection of acute infections with low plasma microbial RNAemia (<10,000 copies/mL)[15].

Our study itself also had limitations. We focused on the RNA fraction and may have missed genomic DNA of other microbes like viruses, bacteria, fungi, and protozoa. Nonetheless, our approach allowed us to identify their transcripts [32]. The identification of *Staphylococcus haemolyticus* and *Candida parapsilosis* could be due to contamination at the time of collection. The use of negative controls indicate that these were less likely from processing in the laboratory. Contamination at time of collection is a common consideration for blood culture testing which needs to be considered as well for sequencing approaches.

## Conclusion

In this study using curated samples, we observed RNA mNGS to be a clinically useful diagnostic approach in a region with diverse causes of infectious diseases. mNGS identified pathogens that would have otherwise been undetected in a protocolized AFI surveillance study in Uganda. mNGS filled current gaps in diagnostic workflows by identifying pathogens not widely known to be circulating in agrarian countries with diverse causes of infection. In addition, our findings further support setting up prospective surveillance for MSF in SSA to allow for early treatment of *R. conorii* infections. Early recognition, empiric antibiotics, and/or

diagnostics could prevent hospitalization or deaths due to rickettsial illness. In addition to demonstrating the diagnostic utility of mNGS, our study highlights the need to identify the burden of rickettsial infections in SSA. These are essential for development of rapid diagnostics, consideration of empiric treatments, and to inform public health policy.

## Declarations

The contents of this article are the sole responsibility of the authors and do not necessarily reflect the views, assertions, opinions, or policies of the Henry M. Jackson Foundation for the Advancement of Military Medicine, Inc., the U.S. Department of Defense, the U.S. government, or any other government or agency. Mention of trade names, commercial products, or organizations does not imply endorsement by the U.S. government. Some of the authors of this work are military service members or employees of the U.S. government. This work was prepared as part of their official duties. Title 17 U.S.C. x105 provides that "Copyright protection under this title is not available for any work of the United States government." Title 17 U.S.C. x101 defines a U.S. government work as a work prepared by a military service member or employee of the U.S. government as part of that person's official duties.

## Supporting information

**S1 Table. Overview of Pavian results.**
(XLSX)

**S2 Table. KrakenUniq read classifications and Pavian Z-scores.**
(XLSX)

## Author Contributions

**Conceptualization:** Abraham J. Kandathil, Paul W. Blair, Matthew L. Robinson, Yukari C. Manabe.

**Data curation:** Abraham J. Kandathil, Paul W. Blair, Jennifer Lu, Raghavendran Anantharam, Kenneth Kobba, Matthew L. Robinson.

**Funding acquisition:** Steven L. Salzberg, Yukari C. Manabe.

**Investigation:** Abraham J. Kandathil, Paul W. Blair, Jennifer Lu, Raghavendran Anantharam, Kenneth Kobba, Matthew L. Robinson, Sultanah Alharthi, Edgar C. Ndawula, Francis Kakooza, Mohammed Lamorde, David L. Thomas.

**Methodology:** Abraham J. Kandathil, Paul W. Blair, Jennifer Lu, Steven L. Salzberg, Yukari C. Manabe.

**Software:** Jennifer Lu, Steven L. Salzberg.

**Supervision:** J. Stephen Dumler, Mohammed Lamorde, David L. Thomas, Steven L. Salzberg, Yukari C. Manabe.

**Writing – original draft:** Abraham J. Kandathil, Paul W. Blair, Jennifer Lu, Matthew L. Robinson, J. Stephen Dumler, David L. Thomas, Steven L. Salzberg, Yukari C. Manabe.

**Writing – review & editing:** Abraham J. Kandathil, Paul W. Blair, Steven L. Salzberg, Yukari C. Manabe.

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
