## [Decision Letter · Decision Letter 0]

24 Apr 2024

Dear Dr. Manabe,

Thank you very much for submitting your manuscript "Metagenomic next generation sequencing of plasma RNA for diagnosis of unexplained, acute febrile illness in Uganda" for consideration at PLOS Neglected Tropical Diseases. As with all papers reviewed by the journal, your manuscript was reviewed by members of the editorial board and by several independent reviewers. The reviewers appreciated the attention to an important topic. Based on the reviews, we are likely to accept this manuscript for publication, providing that you modify the manuscript according to the review recommendations. 

Sincerely,

Gabriel Rinaldi, M.D., Ph.D.

Academic Editor

Elsio Wunder Jr

Section Editor

Reviewer's Responses to Questions

**Key Review Criteria Required for Acceptance?**

**Methods**

-Are the objectives of the study clearly articulated with a clear testable hypothesis stated?

-Is the study design appropriate to address the stated objectives?

-Is the population clearly described and appropriate for the hypothesis being tested?

-Is the sample size sufficient to ensure adequate power to address the hypothesis being tested?

-Were correct statistical analysis used to support conclusions?

-Are there concerns about ethical or regulatory requirements being met?

Reviewer #1: The study design is clear. This is more of an exploratory study based on the generation of large genomics data sets rather than hypothesis testing. The authors aim to evalute the use of RNA metagenomics in diagnosis for patients with unknown illnesses. It should be possible to state the study objectives more clearly as a hypothesis, if this is required. Statistical approaches based on Kraken were used, a well established metagenomics software. 

It could be further clarified ablout the sufficient sequencing depth used to perform this study. Data should be given to summarize the numbers of reads per sample (after mapping to the human genome), and to summarize the z-scores for Pavian in a quantitative manner. Are these z-scores sufficient for the study? In particular it would be useful to provide the Pavian z-scores for the 9 microbes mentioned in Fig 1 and indicate how they differ from the rest. Were other pathogenic microbes detected by the study, albeit with less significant z-scores? It would be useful for the authors to provide a full summary of the data, beyond the highlights of interest here, so the reader can get a more general feel for what the method detected, including any background signals. 

It would probably be helpful to mention the samples were taken from plasma in the abstract.

Reviewer #2: The Objectives of the study are clearly articulated and appropriate. There are no concerns about the ethical conduct of the study per the approvals and adherence standards provided in the Methods. 

The study was appropriate for addressing the stated objective to assess the use-case for mNGS as a epidemiologic & diagnostic tool for AFI in settings with diverse potentially diverse and emerging etiologies. 

For the question being asked, the sample size is sufficient, and adequate power is not a necessary consideration for this descriptive epidemiologic/diagnostic evaluation. 

The use of positive and negative controls, both clinically derived and commercially sourced, is a notable effort for methodologic rigor. The panel-based adjudication process for establishing clinical relevance of an mNGS detection is acceptable approach. 

MAJOR

Lines 65-67: thrombocytopenia & leukopenia thresholds

For the purposes of a reader being able to reconstruct “what was the target patient population?” or “what is the provenance of the participants included in this report?” was there a threshold criterion for thrombocytopenia (e.g., platelets < 100,000?) and was there a threshold criterion for leukopenia (e.g., white blood cell count < 4000?)? 

MINOR 

The study population is appropriate; description of the study population resides in citation 10. To assist the reader’s experience, I recommend the author’s include a simple sentence/clause BRIEFLY stating the target population/eligibility criteria: adults with a ≥ 38.0 °C temperature or history of fever within 48 h of presentation (taken from the Abstract of citation 10). Such a sentence will help the reader decide “did the authors aim to recruit a generalizable patient population for this study?” without having to go to PubMed to dig up citation 10. 

Line 78: extraction protocol to reduce DNA contamination. 

For the sake of clarity & methods reproducibility, was there a step after centrifugation at 1600g x 15 minutes before filtration using syringe filtration? 

In other words, after centrifugation perhaps a certain fraction (the pellet or some other layer) was pipetted and then this fraction was subjected to the additional filtration? 

Lines 88-90: assessment of library quality prior to sequencing

Can the others expand on the criteria used for qualitative assessment of each library by the Bioanalyzer and any criteria used for the quantitative assessment by PCR? For example, was there a minimum quantitative threshold that would have disqualified certain samples from being included on the sequencing run? 

Lines 115-116: k-mer count. 

Can the authors expand on how the “unique k-mer” count from KrakenUniq reports was used? 

At present, it is unclear. A reader could speculate that reads mapping to low complexity regions were considered suspicious for spurious detections rather than true infection…? 

Lines 116-117: read counts and genome coverage

Was there a minimum threshold criterion for number of read counts in order to consider a detection bona fide? 

In some publications, I have seen >50 reads used as a threshold. While this might be semi-arbitrary, it would still be helpful to readers in the field to know any thresholds or decision criteria you used in making calls. 

From Figure 1 one can see that the number of reads that mapped to R conorii was a small number (and in what turned out to be confirmed by PCR and Sanger sequencing). So if there was not a minimum read threshold, it would still be helpful to state this and/or clarify further the interpretive approach that was used. 

The same questions apply to genome coverage: was there an operational target that you used? 

I realize that with the last 3 comments (library quality assessment, k-mer count, and read count threshold / genome coverage threshold) that there might not be standardized criteria (and perhaps things passed “an eyeball test” (How do I know these were real detections? I know it when I see it.) 

So these minor queries are not so much meant to be nit-picky as for the sake of Methods reproducibility that would be of interest to the field and to readers who have interest in making platforms like this clinically relevant and actionable. 

HIV Status: 

HIV infection status is reported in Table 1 of Results, but there was no accompanying explanation in the Methods how this was established: participant self-report? Rapid antibody testing? 

Suggest including a sentence towards the beginning of Methods referring the reader to citation 10 for how the demographic/clinical/physiologic information in Table 1 was collected (assuming that citation 10 covers this data collection).

**Results**

-Does the analysis presented match the analysis plan?

-Are the results clearly and completely presented?

-Are the figures (Tables, Images) of sufficient quality for clarity?

Reviewer #1: The analysis matches the plan. The tables are clear but Figure 1 is of poor quality, and the text is illegible. This needs to be replaced with a higher resolution version.

Reviewer #2: The analysis presented matches the analysis plan, and the results are clearly presented. 

Figure 2 image file is fuzzy, but hopefully the resolution of a final version will be better. 

MINOR

Lines 173-175: relevance of P falciparum sequence detections

It is arguable whether a detection of P falciparum sequences is indeed clinically relevant in a patient who was smear-negative and malaria RDT negative. The clinical relevance of PCR-based detection of P falciparum is likewise problematic. In other words, this is not a criticism of mNGS, but rather nucleic-acid based detection of P falciparum more generally, as most clinicians would assert that clinically significant P falciparum infection typically includes presence on smear. Was there any evidence that the patient was treated for malaria prior to admission/enrollment? 

This detection of P falciparum can still be rendered, in my opinion, as clinically relevant, not so much because malaria would have been the cause of the patient’s illness but because recent or concurrent malaria is associated with non-typhoidal Salmonella (one of the most common causes of meningitis in Africa). I suggest the author’s consider further clarification of this case: whether prior treatment with anti-malarials is known vs unknown and wording that acknowledges it might be problematic to attribute illness to malaria in the context of P falciparum nucleic-acid detection with a negative blood parasite smear.

**Conclusions**

-Are the conclusions supported by the data presented?

-Are the limitations of analysis clearly described?

-Do the authors discuss how these data can be helpful to advance our understanding of the topic under study?

-Is public health relevance addressed?

Reviewer #1: The conclusions are supported by the data presented, although some additional detail could be given (see the methods section). 

Limitations are clearly described in two paragraphs of the discussion.

The data are clearly connected to public health issues and help with the diagnosis of unexplained illness.

Line 239/240: please clarify DNA pathogens and "transcripts of DNA microbes". This is currently non-sensical? Is microbial DNA meant?

Reviewer #2: The conclusions are supported by the data presented. Limitations of mNGS are described. Public health relevance of R conorii detection and the strengths and limitations of mNGS as a surveillance modality are discussed. 

Conclusion about MSF in SSA: 

The phrasing in the Abstract is technically correct “first report to describe the molecular confirmation of a hospitalized case of MSF in SSA.”

However, in the Conclusion, the rendering is slightly more problematic: “first report of patient admitted with molecularly confirmed locally acquired R conorii infection in SSA.”

AJTMH 2005;73:1086 describes an MSF case acquired in Kenya where the patient returned to Japan and was admitted to hospital in Japan. The Parola et al 2013 global review of rickettsioses (Clin Microbiol Rev 2013;26:657) cites the case of a South African man who acquired MSF in South Africa and then presented with a hemorrhagic fever-like while traveling in Brazil. So I think both of these would be cases of MSF acquired in SSA that required hospital admission. 

I recommend the phrasing in the Discussion mirror the phrasing in the Abstract. 

Regardless of these semantics, I do think this genetically confirmed detection of MSF in Uganda is significant, as fatal spotted fever rickettsiosis (no speciation) has been described in Kenya and the detection presented in this study is, to my knowledge, only the second genetically confirmed detection of MSF in East Africa, where many assume that the less virulent R africae / African Tick Bite Fever is the main spotted fever rickettsiosis.

**Editorial and Data Presentation Modifications?**

Reviewer #1: This paper is acceptable with minor revisions.

Reviewer #2: See aforementioned comment about the blurriness of the Figure.

**Summary and General Comments**

Reviewer #1: The authors have applied RNA metagenomics to 42 patients from Uganda wtih unexplained acute febrile illness. The data were analysed using Kraken, a well established computational method designed for this kind of study. The data identify pathogenic organisms that had been missed by other approaches, thus demonstrating the utility of the methodology for diagnosis. Some further detail and summary statistics could be given about the analysis, but otherwise this appears to be a fairly routine application of metagenomics to a proof of concept study. It clearly demonstrates the utility of metagenomics for treating patients with unknown illnesses, and has significance for public health issues and medical diagnosis. 

The data sets are not yet available, although the authors promise they will be made available on publication. It is useful for reviewers to check that the correct details and URLs are provided in the manuscript before publication.

Reviewer #2: Overall, I think this work adds value to the field 1) showing the strengths (unbiased nature) and limitations (diminished sensitivity and overall low detection frequency) of mNGS as a surveillance tool. 2) The detection of R conorii is a public health relevance as this is a neglected disease and severe disease is more common with R conorii than with R africae. 

As stated in my comments above, I think the manuscript would benefit from some additional linkage to the parent study (e.g., providing the inclusion criteria), expanding on some methodologic details (namely for mNGS interpretive methods but also providing details of the selection criteria for thrombocytopenia and leukopenia). 3) I have suggested some nuance around the novelty of the R conorii detection in Uganda.

PLOS authors have the option to publish the peer review history of their article (what does this mean?). If published, this will include your full peer review and any attached files.

Reviewer #1: No

Reviewer #2: No

Figure Files:

Data Requirements:

Reproducibility:

References

---

## [Decision Letter · Decision Letter 1]

17 Jul 2024

Dear Dr. Manabe,

Thank you very much for submitting your manuscript "Metagenomic next generation sequencing of plasma RNA for diagnosis of unexplained, acute febrile illness in Uganda" for consideration at PLOS Neglected Tropical Diseases. As with all papers reviewed by the journal, your manuscript was reviewed by members of the editorial board and by several independent reviewers. The reviewers appreciated the attention to an important topic. Based on the reviews, we are likely to accept this manuscript for publication, providing that you modify the manuscript according to the review recommendations. 

Sincerely,

Gabriel Rinaldi, M.D., Ph.D.

Academic Editor

Elsio Wunder Jr

Section Editor

Reviewer's Responses to Questions

**Key Review Criteria Required for Acceptance?**

**Methods**

-Are the objectives of the study clearly articulated with a clear testable hypothesis stated?

-Is the study design appropriate to address the stated objectives?

-Is the population clearly described and appropriate for the hypothesis being tested?

-Is the sample size sufficient to ensure adequate power to address the hypothesis being tested?

-Were correct statistical analysis used to support conclusions?

-Are there concerns about ethical or regulatory requirements being met?

Reviewer #1: (No Response)

Reviewer #2: (No Response)

**Results**

-Does the analysis presented match the analysis plan?

-Are the results clearly and completely presented?

-Are the figures (Tables, Images) of sufficient quality for clarity?

Reviewer #1: (No Response)

Reviewer #2: (No Response)

**Conclusions**

-Are the conclusions supported by the data presented?

-Are the limitations of analysis clearly described?

-Do the authors discuss how these data can be helpful to advance our understanding of the topic under study?

-Is public health relevance addressed?

Reviewer #1: (No Response)

Reviewer #2: (No Response)

**Editorial and Data Presentation Modifications?**

Reviewer #1: (No Response)

Reviewer #2: (No Response)

**Summary and General Comments**

Reviewer #1: The authors have mainly addressed my previous comments well and the paper is very nearly suitable for publcation. However, I do have a couple of very minor issues. 

The authors have left the phrase "DNA microbes" in the paper. I find this phrase confusing. It is clear from their comment that they mean DNA viruses - which is perfectly sensible. But do they intend to include other microbes? It surely makes no sense to talk about DNA bacteria for instance. Would it be clearer to change "DNA microbes" to "DNA viruses"? The paper they now cite from Kumata et al only appears to refer to DNA viruses, I could not find "DNA microbes" in that paper. However, I will leave this issue to the authors discretion. 

The authors state that the data have been uploaded under project ID PRJNA1115716, but this ID gives no hits on the SRA or NCBI. Perhaps it is incorrect or not yet released? The PLOS Data policy requires authors to make all data underlying the findings described in their manuscript fully available without restriction, and so I assume the paper should not be published until this has been sorted out.

Reviewer #2: The authors have responded adequately to my concerns and suggestions. Of note, the line numbering in their responses did not align with the revised manuscript version I received, so it did make the process of reviewing their responses more onerous than one would have hoped. Overall, though, their additions and edits have improved the manuscript and increased its informativeness.

PLOS authors have the option to publish the peer review history of their article (what does this mean?). If published, this will include your full peer review and any attached files.

Reviewer #1: No

Reviewer #2: No

Figure Files:

Data Requirements:

Reproducibility:

References

---

## [Editor Report · Decision Letter 2]

11 Aug 2024

Dear Dr. Manabe,

We are pleased to inform you that your manuscript 'Metagenomic next generation sequencing of plasma RNA for diagnosis of unexplained, acute febrile illness in Uganda' has been provisionally accepted for publication in PLOS Neglected Tropical Diseases.

Best regards,

Gabriel Rinaldi, M.D., Ph.D.

Academic Editor

Elsio Wunder Jr

Section Editor

---

## [Editor Report · Acceptance letter]

13 Sep 2024

Dear Dr. Manabe,

We are delighted to inform you that your manuscript, "Metagenomic next generation sequencing of plasma RNA for diagnosis of unexplained, acute febrile illness in Uganda," has been formally accepted for publication in PLOS Neglected Tropical Diseases.

Best regards,

Shaden Kamhawi

co-Editor-in-Chief

Paul Brindley

co-Editor-in-Chief
